# Bracing Ourselves: Embracing Policy Changes through a Long-Standing University–Schools Teacher Education Partnership in England

Nicola Warren-Lee , Lorna Smith, Janet Orchard *, Lucy Kelly, Jon James and Alf Coles

School of Education, University of Bristol, Bristol BS8 1QU, UK; nicola.warren-lee@bristol.ac.uk (N.W.-L.);
lorna.smith@bristol.ac.uk (L.S.); lucy.kelly@bristol.ac.uk (L.K.); jon.james@bristol.ac.uk (J.J.);
alf.coles@bristol.ac.uk (A.C.)
* Correspondence: edjlo@bristol.ac.uk

**Abstract:** We articulate the principles behind our practices as teacher educators for an Initial Teacher Education (ITE) provider in South-West England working in partnership with local schools. We position them as 'braces' that enable us not only to withstand but 'embrace' the challenges of the current policy environment, which imposes braces of its own. All ITE providers in England must present their curriculum and teaching plans for approval by the Department for Education, one of many significant changes to ITE required in England over the last decade. Furthermore, for such plans to be approved they must assume a model of schooling based on knowledge acquisition in tension with not only our own motivations but also a broader (global) conversation among teacher educators critical of policy priorities for schooling and professional teacher formation that prioritise success in high-stakes tests and international league tables. The braces/principles emerge through articulating our collective understandings of and concerns with the policy braces. Our principles connect to the processes we feel confident are necessary aspects of 'becoming' teachers and can hold our Post Graduate Certificate in Education (PGCE) together, going forwards. The braces—becoming an informed educator; becoming an ethical actor; becoming a reflective practitioner—have a distinctive form of reflexivity at their heart and should apply to everyone across our partnership: us, as university-based teacher educators; school-based colleagues; and pre-service teachers. We offer our approach and the braces/principles we have developed to others also struggling to reconcile their own notions of good teacher education with national policy imperatives.

**Keywords:** initial teacher education; ITE policy; principled practice

## 1. Introduction

Each of the authors of this article works on an ITE programme at the University of Bristol in the South-West of England. We have written it together to identify, articulate and advocate for some common principles guiding our work and our course design. The historical moment in which we have collaborated on this work is significant (2023): the last decade has seen a significant rise in the influence of the Department for Education (DfE) in England over teacher education, culminating in the requirement for ITE providers to produce policy documents which seek to control, for the first time, how teacher education is 'done'. We share our responses to those recent DfE requirements in light of our collective experiences and understandings of effective teacher education, situating the challenges we face within the broad political and historical landscape of ITE in England. Reflecting together on a series of tensions created by 'braces' on our practice imposed by policy makers, we have generated a set of braces/principles that help us and may be of interest to others facing similar challenges.

The first tension is how to navigate 'top-down' initiatives, imposed from above and outside our partnership, that we do not 'own'. We review existing research into the known

difficulties that such approaches provoke for teachers, leading us to assert the value of the principle of 'becoming a reflective practitioner'. Secondly, we consider tensions around navigating 'research-based' findings or prescriptions about teaching—particularly when these appear politically driven—leading us to assert the principle of 'become an informed educator'. Finally, we recognise more general tensions teachers face as moral agents experiencing conflict between what they are required to do and their educational value assumptions, leading us to highlight the principle of 'becoming an ethical actor'. Drawing these together, we brace ourselves for challenges by embracing these three principles of teachers' 'becoming' as a non-deficit model of teacher learning, which, for us, is non-negotiable.

Let us briefly say more about the terms 'embrace' and 'brace' used herein. The word *brace* derives from the French *bras*, from the Latin *brachium*, 'an arm'. The Oxford English Dictionary indicates that *brace* was first used metaphorically in English around 1400 to refer to an 'arm' of water. It was adopted more literally later to suggest the encircling of arms around something (or someone), as in today's 'embrace'. But, arms have many uses, and so brace too became polysemous. Nautically, a brace (e.g., "splice the mainbrace"), like a rigid arm, is a support. This gives brace positive connotations: it provides strength and structure. Yet, the association with rigidity and strength gradually led to brace being used as a verb more figuratively to indicate defiance ("to bluster, domineer" c. 1447) or an inappropriate use of force (as in the US slang, "to charge extortionately"), suggesting attendant connotations of danger which one might have to brace oneself to withstand.

The ambiguities of 'brace' are reflected in this article. Imposed braces—structures externally imposed upon ITE in England—are in tension with the principles that we see as underpinning (or bracing) our own programme. We require collective rigidity and strength to brace ourselves to withstand threats to the programme (as we see them). This comes from a shared commitment to our three key principles (becoming an informed educator; becoming an ethical actor; becoming a reflective practitioner) which enable us to embrace both the pre-service teachers with whom we work and our ITE partnership to support the development of humane, creative, reflective, and reflexive practitioners. As the three key principles embrace us and support us, we can go about our work with greater confidence.

## 2. Broad Landscape of Initial Teacher Education in England

In the United Kingdom, responsibility for education—including teacher education—has been devolved across the four nations (England, Scotland, Northern Ireland, and Wales) for several decades. We are concerned specifically with ITE policy reforms in England, which has seen particularly intensive government intervention, backed by an ideological commitment to moving ITE from being predominantly positioned within higher education institutions (HEIs) to being situated in and administered by schools [1]. Learning to teach originally took place through learning from experience 'on the job' [2]. More formalised training for teachers developed in piecemeal fashion from the nineteenth century onwards, with two main course structures emerging as routes into teaching: a three-year undergraduate degree in any subject, followed by a one-year course of professional training (the model our own practice continues to follow and overwhelmingly the most popular route overall into teaching), or a degree combining the study of education and professional training.

One long-standing concern with university-based teacher education has been that pre-service teachers may be too preoccupied with practical concerns to attend much to formal academic theory-driven lectures and seminars [3]. Hence, practice-orientated seminars, for example, on teaching and learning, and managing behaviour in classrooms are well-established elements of PGCE programmes like ours, the emphasis being on partnership between schools and universities, a dialectic relationship between theory and practice in teacher education firmly established. However, such programmes have not been met with universal approval, and a narrowing of the perception of the teacher has taken place, from the teacher as an autonomous curriculum developer to an "executive technician" [4]. This

has undermined and marginalised those aspects of professional programmes that, like ours, aim to introduce teachers to the wider horizons of their work [5].

There are now many school-based routes through which beginning teachers can gain Qualified Teacher Status (QTS), something which distinguishes ITE in England from that seen in other global contexts [6]. While universities in England are often still involved in these school-based programmes through the provision of an academic component enabling pre-service teachers to attain the PGCE at Master's level, a clear shift can be seen in how ITE is framed. Rather than a scholarly autonomous process of learning, teacher education is now popularly construed as a service or product [7], with HEI institutions like ours referred to as "providers".

So, while many countries have experienced reforms in their teacher education systems resulting in shifts to more instrumentalist approaches [8], England has seen a distinctive government ideology enacted that construes teacher education as something to be reformed and marketised [9]. This article might serve as something of a warning for the practice of teacher education when marketisation takes hold. The DfE recently joined forces with the Department for International Trade (DIT) to produce an international education strategy that overtly packages teacher education as a means to promote recovery and economic growth for the UK post pandemic and post Brexit [10]. The strategy aims to "increase the value of our education exports to £35 billion per year" (p. 5), instigating a new qualification, the International Qualified Teacher Status (iQTS), which provides an opportunity for students globally to undertake teacher education programmes subject to English ITE standards.

Similarly, the DfE's recent review of domestic ITE, dubbed the "Market Review" [11], is also reframed towards a market, employing language which presents the process of learning to teach as a commodity which can be made more efficient and of a higher quality. This requires any ITE institution that wants to gain accreditation, whether university- or school-based, to buy into the delivery of a Core Content Framework [12] prescribing what and how pre-service teachers should learn, how placements in schools should be organised, and how mentors should be trained. This aligns with another policy directive, the Early Career Framework, that outlines a programme of professional development for teachers in their first two years of work [13].

Both frameworks make us uneasy, appearing to standardise teacher learning excessively, implying that there is only one way to be pedagogical in the classroom, predicated on "what works" [14]. Learning to teach is presented as a straightforward technical process, based on a systematic knowledge base prescribed by the DfE [15]. Research evidence is drawn upon unproblematically, suggesting that certain teaching actions and strategies will offer guarantees for student achievement. There seems little recognition of the context-sensitive messy nature of teacher learning or the difficulty in ascertaining causal connections between the complexities of teaching and learning [16,17].

As teacher educators, it concerns us that these documents—dependent on a restricted range of evidence—concede little to empowering teacher learning or promoting teacher agency, commitments which we hold dear. Instead, they prescribe pedagogical methods, a tight brace, we suggest, promoted as standard deficit approaches that ignore the pathic [18] and complex nature of teacher practice. Worse, while traditional deficit models of teacher development assume that a teacher's practice can be changed by others transmitting information and ideas to them which they can apply in their classroom settings [19], such changes, in practice, do not readily happen, with teachers struggling to implement new ideas and approaches, feeling that they lack agency and autonomy [20]. Such deficit models focus too often on what a teacher *is not* rather than *what is* (their being) or *what might* be (their becoming). They rarely provide an extensive account of practice that recognises the complexity of teacher development or make room for ambiguity, innovation, or consideration of the context in which beginning teachers learn to teach. The documentation is silent, for example, on the climate crisis, the uncertain impacts of which provide opportunities for

teachers to reflect on the complex and problematic contexts in which they are learning to teach [21].

So, what are we to do? Clarke and Hollingsworth [22] helpfully present notions of teacher change across a spectrum of possible perspectives, ranging from the deficit accounts just described to non-deficit accounts of the kind we endorse, which present change as growth or learning. Non-deficit models recognise the active involvement of teachers in the development of their practice, positioning them as active learners who change through opportunities for professional learning [23,24] in which they are encouraged to undertake critical reflection. Being aware of Clarke and Hollingsworth's generous conception of teacher change, recognising a wide range of possible accounts of teacher change as being nevertheless legitimate, does help us to rationalise our conflicted feelings when we are required to embrace the Early Career and Core Content Framework documentation, while remaining committed to our alternative vision of teacher change and development. Captured by the discourse, we must engage with it constructively, for the sake of the programme and our partnership. So, having sketched the broad landscape and our concerns, we next focus specifically on three key tensions before elaborating a non-negotiable principle as a practical 'constructive as possible' response. The principles overlap deliberately (see Figure 1).

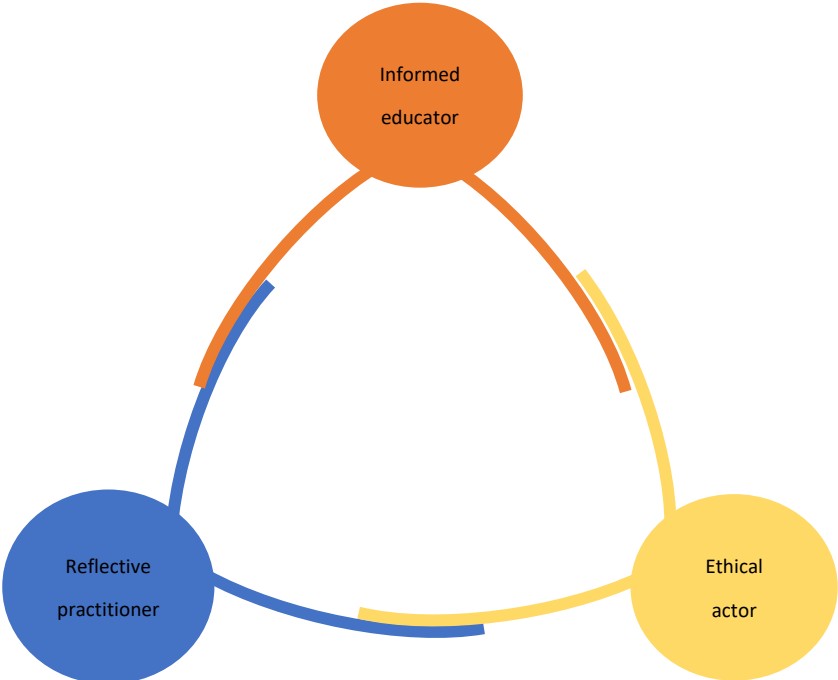

**Figure 1.** The embrace and brace of the three principles of the Bristol course.

## 3. Embracing Top-Down Policy Initiatives—Becoming a Reflective Practitioner

Our first key concern is that current education policy is top-down and accountability-led. How do we encourage pre-service teachers to reflect on their own practice, as we believe they should, in this circumstance? A practitioner's relationship with policy is complex—they must balance being the "actor and object and subject" [25]. They are the subject, because policy is about them, the object, because the policy is directed to them, and the actor, because it is their role to bring it to life. Yet, as Winch et al. [4] imply, when that policy is centrally imposed and inflexible, almost by definition, practitioner agency is limited, impeding reflective practice.

Curriculum control has also been evident in relation to the national curriculum where, for example, the requirements have moved from what "Pupils *should be able* to [do]..." (e.g., 2007 English National Curriculum [26], our emphasis)—wording which implies a facilitatory teacher role—to what they "should be taught" (e.g., 2014 English National

Curriculum [27]), which determines a prescriptive teacher role. This subjugation is exacerbated by a hierarchical accountability system. Education in England takes place "within a dominant audit culture" [28]. Such is the pressure of ensuring that students achieve good examination grades, it is now "a given" [29] in many schools (both primary and secondary) that examination preparation explicitly drives lessons. Increasingly, teachers are required to use pre-planned packages of resources in an attempt by school managers to control teaching quality. Prescription has extended to some schools creating lesson resources to be marketed to others (e.g., see https://www.thenational.academy (accessed on 13 December 2023). By contrast, our programme encourages teachers to aspire to being autonomous curriculum developers, despite being situated in an often-prescriptive context.

While the easy availability of resources apparently reduces teachers' workload, a one-size-fits-all approach is profoundly at odds with Dewey's contention that good teaching is built on interaction between a child's experience and the knowledge being taught [30]. When lessons become "overdetermin[ed]. . . depersonalised closed systems" [31], there is little room for teachers (let alone pre-service teachers) to act agentively, take the initiative, be creative, and respond to the needs of their individual students—all of which we see as fundamental traits of a professional educator [32]. Instead, many teachers feel overwhelmed with "necessity and responsibility" [25], the freedom to make professional decisions in their own classrooms "stifled" [33] by the brace of accountability.

We are not so naïve as to expect that pre-service teachers on our programme will be able to overturn such prescriptive policy in their schools, let alone national policy. However, by encouraging them to become reflective practitioners within the constraints of the model required by policy, we remain true to our principles, hoping that they will develop the necessary agency over time to make small but creative and impactful 'bottom-up' decisions responsive to the individuals—and their associated learning needs—that they encounter in their classes.

Rather than expecting direction, as reflective practitioners they will continue to learn as freely as possible from the work of other educators (an element of becoming an informed educator, which will be discussed further below), also engaging systematically through a cyclical reflection process on the impact of their own teaching and its impact on students. This involves thinking independently in advance of the lesson about the plan and resources to be used, taking into consideration their own strengths, affordances, and any limitations in light of their knowledge of students' needs; it involves keen attention *during* the lesson and introducing adaptations where necessary; it also involves reviewing the lesson afterwards, specifically how the plan and resources were able (or not) to foster learning, informing the planning of the next one.

Learning from critical reflection on experience in the classroom is challenging, implying that teachers are capable of reviewing thoughtfully and systematically what they have implemented in the past with a view to sustaining or improving their practice in the future [4], a behaviour which they are entrusted to implement, in contrast to more tightly controlled and directive approaches to teacher development. At least three types of professional learning are established in the literature, all of which are used in our programme: reflection-in-action; the exercise of scholarship; teachers' undertaking systematic enquiry themselves upon which to base future action (see [4], 204-6) for an extended account of these three aspects).

So, after Schön [34], we will continue to follow structured cycles of practice in our programme, i.e., reflection, during and after practice, recursive reflection on (further) practice embedded in our lesson observation proformas, and both professional studies and curriculum subject-based seminars, while also delivering top-down mandated expectations. Pre-service teachers will continue to be encouraged to engage with selected readings, policy documents, and official recommendations and consider how these impact issues in classroom practice that concern and affect them. We will continue to promote "systematic self-study" as well as "the study of the work of other teachers" and the "testing of theory

in practice", with support from us as specialist education researchers [35], inspired by the notion of teaching as a form of "extended professionalism" advocated by Hoyle [36].

These various forms of reflexivity demand that time be taken during ITE to think and talk through experiences, facilitated by teacher educators (whether in school or university) with the necessary insight, experience, and skill to facilitate thinking processes which enable pre-service teachers to learn to be reflective and reflexive. Nowhere does the model we must follow acknowledge the importance of teachers being able to recognise critical incidents, let alone attend to the strong emotional hues likely to indicate something significant in their practice to work on. Whether the teacher educator structures a short exchange between students or provokes a 'lightbulb moment' which highlights a broader issue to which it may relate, these are the stimuli from which teachers learn to plan appropriate specific actions [37] by way of a response. Often such actions take the form of a conjecture— something to try out—that, in turn, can be reflected upon and adapted. Those pre-service teachers who understand why we take this line, hopefully, with increasing confidence and agency, will develop dispositions to make professional decisions not only after the event—reflection *on* action—but also act in the moment (reflection *in* action [34]), engaging with the deficit model of teacher development critically. There is a risk others may be confused. However, unless we were to capitulate entirely to the braces being imposed on our programme, how are we to sustain the model of reflective (and reflexive) practice in which we and our partnership are invested?

## 4. Embracing Research-Based Prescriptions—Becoming an Informed Educator

We have recently joined others (including the Universities' Council for the Education of Teachers (UCET) and the University of Cambridge) in criticising the recent ITE reforms mentioned above on the grounds that they promote "identikit" teachers [38], de-professionalise teachers' work, and "marginalise long-standing traditions of educational thought" [15]. By contrast, teachers who reflect upon their teaching along the lines just sketched, with a critical research-informed lens, are better able to make transformational changes to improve their own practice and the lives of their students [39,40]. Such reflexivity is a central brace of our work as teacher educators, including the focus here, on becoming informed educators.

The Core Content Framework [12], as Brooks and Hordern observe, represents a "decontextualised series of interventions with narrow objectives" [15]. Yet, in practice, teachers must think broadly through the particularities of a myriad of individual situations daily, acting contingently to create the best course of action as they come to see it. Toolkits provide standard solutions, yet the teacher's challenge is to "situate knowledge in the living context" [41]. Current ITE reforms do not sufficiently acknowledge the critical research-informed thinking and dynamic decision making that pre-service teachers need to develop from the very start of a teacher education course [42,43]. Instead, they act as a rigid structure or brace, focusing only on the content and technical skills to be rehearsed.

Winch et al. [4] identify three distinct aspects of teachers' professional knowledge— situated/tacit/intuitive, 'know how', and 'critical reflection'—found in the wider educational literature concerned with articulating the professional knowledge teachers need, which we consider necessary to being appropriately 'informed'. Combined, these contribute to the 'good sense' teachers need to exercise sound professional judgement when making high-quality context-driven decisions. This is not to suggest that teachers' decision making can be random, nor do we accept that, just because teachers' work is context-specific, any decision/action would be as good as another. Rather, such judgements need to command authority and respect to be effective; hence, informed educators need access to wide sources of evidence in their learning and professional work to support them in making decisions.

One interpretation of how teachers become informed concerns "practical theorising", pioneered by Donald McIntyre. This notion underpins the widely regarded Oxford Internship programme, whereby different sources of knowledge are used in classroom decision making. On this approach, "where irrefutable, generalisable answers are not available, a

process of argument testing occurs—[pre-service] teachers developing reasoned beliefs about what is more or less likely to work" [44]. As part of a practical theorising process, educational theory and research is utilised, along with other forms of evidence, such as colleague feedback, to make sense of experience and inform future activity. Distinctively, on a practical theorising approach, no one source of knowledge is elevated or idealised, and change as well as contextual factors will affect, in any one situation, what is both appropriate and possible.

Teacher education on a non-deficit model broadly uses research evidence within professional learning to reduce reliance on untested personal theories or on marketised temporary approaches; it seeks to develop critically informed educators better able to question and see alternatives to and different perspectives on existing practices. The notion that the "use of evidence, enquiry and evaluation lies at the very heart of what it means to be both a teacher and teacher educator" [45] is broadly supported, but teachers need to learn to be discriminating when using educational research as evidence, rather than straightforwardly aiming to manifest research *into* practice. This requires a dynamic process of constant enquiry and renewal within teacher education; becoming an informed educator is a "conditional, tentative and quite complex" [46] process.

In supporting pre-service teachers to become informed educators, we recognise that they start out with preconceptions about the usefulness of research in their teaching practice; moreover, we recognise that engaging with research may elicit an emotional response [47]. In a non-deficit model, one which resists the view of teachers as "executive technicians" [4], teacher educators take time and invest energy and skills in addressing these factors known to affect pre-service teachers' research engagement. Fully supporting the notion that "research and enquiry have a major contribution to make to effective teacher education" [48] and, positioned together, that they are crucial for continual professional renewal [49], once again we must be creative in responding to the braces being imposed by a deficient, reductive, and narrow understanding of teachers' research engagement.

Let us take as an example research based on the potential of neuroscience to influence our understandings of the learning process, currently including the "Science of Learning" or "Science of Education" [50,51], an influential research area used increasingly to drive policy in England. As well as knowing research results, informed teachers must develop the confidence and skills to evaluate claims made about teaching and learning based on neuroscience. In a fast-moving field, we want pre-service teachers, as informed educators, to consider critically the relevance and applicability of such ideas to their subject. Reflexivity is critical in engaging with research results. We do not suggest that pre-service teachers must enter the profession enacting the science of learning theories, recognising their influence, but rather to consider thoughtfully what they might mean for their classrooms, allowing for the possibility that it might not be anything.

## 5. Embracing Certainty in the Face of Legitimately Different Beliefs—Becoming an Ethical Actor

Our qualified responses to including the "Science of Learning" in teacher education curricula leads us neatly to the final tension we wish to discuss, the ethical dimensions of teaching. We understand ethics as a praxis that is "morally committed, and oriented and informed by traditions in a field" [52]. Although intellectual thought lies at the heart of what it means to become an informed educator, such knowledge claims are contested. On a non-deficit account of teacher development, teachers need to learn to make sense of varied and sometimes conflicting and yet reasonable and legitimate understandings of education and being a teacher. Moreover, teaching is an inherently relational practice, "embodied, played out in specific social-cultural contexts'" which are "changing over the course of a career for reasons beyond the control of any teacher" [53]. Unique ethically complex situations arise to which teachers need to respond, ranging from 'big' concerns with whether what they are teaching/how they teach is 'good' or worthwhile to 'small' practical everyday questions like whether they should silence children in classrooms [54].

Becoming an ethical actor concerns teachers learning to articulate and defend their own ideals. Even though they recognise that ideal ends may never be realised and that their own influence is limited, "they hold on to the ideal aims, because they believe these ideals best express what is in the interest of children" [55], reinforcing again the need to balance self-interest in teaching with the interests of learners [56], recognising the importance of retaining a connection to broader educational ideals in teaching goes further to connect questions of why teachers teach as they do to more fundamental questions about the lives teachers lead. He observes that this compelling link is "what draws us to the practice of teaching and what sustains us there in the face of difficulty" [57]. To become ethical in this sense alludes to the responsibility teachers have for self-care, an entitlement to flourish themselves as teachers, alongside caring for others.

Simply knowing the accepted ethical conventions, being able to reproduce these accurately and reliably, constitutes an unhelpfully thin conception of becoming ethical as a teacher. Rather, good classroom practice depends on teachers' professional judgement, the capacity to weigh up what might be the best action in specific circumstances [28]. In "*Should ought be taught*", Mahony [58] raises the important question of whether teachers can be educated to acquire ethical judgement or whether such behaviour is acquired 'naturally'. Professional judgement is *developed*, an insight we carefully factor in. Pre-service teachers 'ought' to have the time and space to think, we maintain, to reflect on their evolving professional identity; we see 'thinking' and embracing well-considered values and assumptions as teachers as inter-connected. Generally, though, insufficient time and space is dedicated currently in ITE to the moral matters of value and opinion on which this rests [59].

Yet, paying attention to the ethical, we contend, is also non-negotiable: pre-service teachers (and teachers more widely) need space for "noticing and reacting" [60]. Whilst pre-service teachers might not be able to overturn prescriptive policy in school, in our PGCE programme they can question it, noticing gaps, perhaps, and absences. With becoming ethical continuing to be prioritised, pre-service teachers are supported to think through small bottom-up decisions as noted earlier, leading to increased levels of confidence, agency, and creativity. Being encouraged to 'notice' enables thinking about those aspects of education that cannot be measured (including the braces of accountability and performativity detailed earlier), opportunities to develop ongoing and renewed understandings of what it means to be a teacher. We support teachers in managing the balancing act of being the "actor and object and subject" [60], assuming this to still be possible. 'Noticing' should also help pre-service teachers to reflect critically, i.e., autonomously, across the spectrum of possible accounts of a teacher, from the "deliverers of [. . .] results" [28] and knowledge enshrined in the ECF and CCF to our informed, ethical, and reflective conception.

Current ethical work on our PGCE course includes strands to prompt pre-service teachers' awareness towards decolonising the curriculum and consider the role of sustainability and global challenges in their subject's teaching [21]. Both aspects inform our new curriculum design, although they are extraneous to what policy makers require of us. We recognise the inner work required to uncover biases and power dynamics that can be embedded (and hence hidden) in societal assumptions (e.g., about what it means to be a 'good' child at school). Here, reflexivity about our own reactions and what they tell us about our assumptions is key, acknowledging the "ethical responsibility" of teaching [61]. Our commitment to 'noticing' and 'reacting' relates closely to our more general commitment to reflective practice, which enables the "textured nature" [4] of teaching to be foregrounded and celebrated.

Thus, we accommodate our belief that teachers must navigate complexity and disagreement within a top-down system that offers directives to be operationalised, not critiqued. Locally, we continue to encourage pre-service teachers to make sense of their experiences with the possibility of moral agency and privileged understanding "of the personal, professional and wider social contexts within which they, as practitioners, operate" [62]. While it is demanding, the artistry we consider key to making a career in

teaching fulfilling, "open-minded", "inquiring", and "professional" [63] must be squeezed in, allowing pre-service teachers to still be able to question and consider their ongoing relationship with policy in our programme, such that it "allows a repositioning of the teacher as a person" [60].

Once again, commitment to the "humanising" [64] nature of teacher education is, for us, non-negotiable. It enables teachers to hold their "value base" [63] up to their practice and to notice whether or not there is alignment with their own considered commitments. Through immersion into in-depth critical reflective practice, we seek to embrace pre-service teachers' need to "make sense of their practice by helping them to broaden their perspectives" [65], insisting that they have a space in which to develop such independent judgement.

## 6. Principles of the University of Bristol PGCE Course

In the sections above, we have elaborated three principles that articulate and structure our teacher education course: becoming a reflective practitioner; becoming an informed educator; and becoming an ethical actor. We have tried to show the connections between these principles, which are, in part, an articulation of a logic of practice which has evolved over the last 30 years in the ITE course at the University of Bristol. Our PGCE course was designed in a principled manner, and, as we have shown, teacher reflexivity is a common theme in our practices. Anecdotally, this quality is what is most valued in the partnership of schools in which many of our pre-service teachers work. In becoming a reflective practitioner, embracing reflexivity is required in relation to one's own actions; in becoming an informed educator, reflexivity is manifested through evaluating the claims of others; in becoming an ethical actor, reflexivity within professional relationships is fundamental.

We visualise the three principles and their inter-relationships in Figure 1. The principles form an embrace, supporting the development of pre-service teachers and teacher educators. The strength of the triangle is also present in the combination of principles, offering a brace against unwanted policy pressures. We visualise pre-service teachers and teacher educators as inhabiting and navigating the space created in-between the embrace/brace of the three principles.

To reiterate, we—as a group of teacher educators—have been challenged to articulate the principles behind our practices in response to the DfE's "Market Review". These are the principles by which we have planned our new course (for first teaching in 2024–2025) in a manner that connects to our history (and the thinking reviewed in earlier sections) while also meeting the DfE demands of adhering to the Core Curriculum Framework for ITE and compliance with the Early Career Framework. The principles offer us, as teacher educators, a brace against the pressure of centralisation.

We conceptualise learning to teach as a process of embracing our three principles. The three principles aim to respect the complexity of becoming a teacher. The reflexivity that is within each one also applies to *our* work with those principles. In other words, the three 'becomings' are ones we recognise in ourselves, as teacher educators. Each element of reflexivity can apply to our own practice of teacher education as much as it can support our pre-service teachers in their practice of teaching. For example, implications from the "Science of Education" are ones we do not simply aim to tell pre-service teachers about but rather ones we aim to critically enact. (For instance, rather than offer lectures about the importance of curiosity, we set up situations where pre-service teachers can exercise their own curiosity in considering the role of curiosity in learning, e.g., by considering a range of theorisations of learning.) The curriculum for teacher education that we have devised is not fixed, but itself is in a process of becoming, as we reflect on our actions and reactions and their implications.

## 7. Conclusions

In this article, we have articulated the principles behind our practice as teacher educators to model how we have navigated policy changes we find challenging in our context.

We conceptualise teacher education as centred on three interwoven principles, or processes of 'becoming' informed educators, ethical actors, and reflective practitioners, and we offer to others, similarly tasked with reconciling locally held values with national imperatives, both the model we have evolved and the process by which we have been able to embrace potential constraints by moulding them into principles/braces that continue to support what we do. Our teacher education course aims to provide pre-service teachers with the opportunity to embrace these principles and open themselves up to processes of becoming where the end point of that becoming is not known (and, indeed, is never reached). We recognise that this is challenging work that requires a willingness to engage in processes that involve vulnerability. We aim to engage in such processes ourselves, as teacher educators, and to create a place of emotional safety at the University, where the vicissitudes of those becomings can be shared and explored in a supportive manner.

We conceptualise becoming a teacher as a living practice with no fixed destination, which requires fixed constraints to be navigated (e.g., being able to teach the current curriculum in school, adhering to current demands around assessment of students and inspection of schools). Our principles encompass the Core Content Framework [12], despite the fact that its articulation of teaching in knowledge statements are ones we would not choose ourselves, if we had the option. Rather, we support pre-service teachers on our course to demonstrate that they are engaged in learning pertaining to each statement, while also focusing on the broader and on-going processes of becoming a teacher, which do not end with the conclusion of our PGCE course. In essence, what we have articulated is an embodied perspective on becoming a teacher, recognising that what individuals bring to the profession changes what it means to be a teacher, i.e., an individual navigating a unique journey of becoming which extends throughout their career.

Through engaging in our work with our local school partners and with our pre-service teachers, conceptualised as processes of becoming, we quite deliberately aim to de-centre power away from ourselves, as providers of teacher education. We place pre-service teachers as 'teachers' and 'researchers' from the start of the course (for instance, getting them into school and working with children within the first few weeks). We do not separate theory from practice and engage in dialogic and on-going relationships with our school partners; we view the 'knowings' articulated by school mentors in the same way as we view the articulations of 'knowings' in academic research. In both cases, our pre-service teachers (and ourselves as teacher educators) need to engage in a critical and reflective dialogue with what is offered, testing out ideas in classrooms and being open to change while also gaining a growing sense of who we are becoming as teachers and so what may be possible in our own teaching.

As we hope to have shown, we view reflexivity as central to the becoming of a teacher. We also view reflexivity as a necessary skill for navigating a changing political landscape. As teacher educators, we aim both to embody and occasion a creative and, if necessary, subversive approach to meeting central demands (e.g., from a government). There are times, as teachers and teacher educators, when demands simply need to be met. Through engaging in processes of becoming, we offer an image of how a seemingly rigid system of braces (e.g., which sets out knowledge as inert) can be embraced in a playful and imaginative yet principled and determined manner, satisfying external requirements while opening oneself to becoming ever more reflective, informed, and ethical.

**Author Contributions:** Conceptualization, N.W.-L., L.S., J.O., L.K., J.J. and A.C.; investigation, N.W.-L., L.S., J.O., L.K., J.J. and A.C.; resources, N.W.-L., L.S., J.O., L.K., J.J. and A.C.; writing—original draft preparation, N.W.-L., L.S., J.O., L.K., J.J. and A.C.; writing—review and editing, N.W.-L., L.S., J.O., L.K., J.J. and A.C.; project administration, N.W.-L., L.S., J.O., L.K., J.J. and A.C. All authors have read and agreed to the published version of the manuscript.

**Funding:** This research received no external funding.

**Institutional Review Board Statement:** Not applicable.

**Informed Consent Statement:** Not applicable.

**Data Availability Statement:** The original contributions presented in the study are included in the article, further inquiries can be directed to the corresponding author.

**Conflicts of Interest:** The authors declare no conflicts of interest.

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
