# Peer review of "Bracing Ourselves: Embracing Policy Changes through a Long-Standing University–Schools Teacher Education Partnership in England"

_education, doi:10.3390/educsci14020158_

Round 1

Reviewer 1 Report

Comments and Suggestions for Authors

Thank you for the opportunity to review this timely paper, which sets out the way in which one university initial teacher education (ITE) provider in England has developed its PGCE programme in response to recent government requirements and in accordance with a set of programme-wide principles. While much of the argument will resonate with university providers of ITE in England, in particular, it could be more strongly theorised overall in order to increase its significance as a paper.

The paper begins with an historical overview of teacher education provision in England, although this is, by necessity, partial and covers much of the ground previously covered by other authors who have tracked a similar trajectory.  The possible danger of such a broad overview is that much of the nuance and complexity is lost, and so there could be less detail of some of the policy landscape over time, and more emphasis instead on some of the key underlying tensions that have characterised ITE in England over the past four decades (namely the fairly ubiquitous notion of a theory/practice divide; the related university-led/school-led discourse; the contested role of educational research; or various views as to the nature of teacher professionalism). The Winch et al. (2016) Oxford Review paper that followed the BERA Inquiry, and which is appropriately cited in this paper, then becomes an effective starting point and could, as the way in to the central argument, make use of their three conceptualisations of the teacher. This would then link to the Hordern & Brookes (2003) analysis and situate the paper firmly within the current context. It would also help to structure the sections further on in the paper dealing with the ‘three issues’ which could be more tightly linked to this central theme.

The section on teachers’ concerns with ‘top-down’ initiatives does not seem to fit that well with the central argument – yes, of course, the paper is about preparing teachers for a profession in which these initiatives have been only too evident, but this section needs to tie in more closely with the central premise of the paper. Likewise, the section on research-informed teaching seems to wander away from its key point and ends up being a relatively superficial critique of the Core Content Framework, merged with a critique of the Market Review process without being forensic enough about the links between the two. Incidentally, on page 7, the paragraph at the end of the section appears to be more or less a repetition of the previous paragraph.

The paper then moves on to present the ‘principles’ which appear to be embodied in the three ‘becomings’ (which are then discussed as entwined ‘strands’ within a ‘braid’ using the metaphor of a rope). While each of these ‘strands’ or is discussed there is not a strong sense of these being ‘principles’. Rather, the principles seem to emerge more in the penultimate paragraph of the paper (page 11, lines 477-489). The three paragraphs outlining the ‘becomings’ do not seem to follow on logically from what ahs gone before and need to do more to problematise the nature of each of these areas. Why, for example, is ‘Becoming an informed educator’ equated with the need to be versed in the ‘Science of Learning’ approach? Why is ‘Becoming an ethical actor’ focussed on issues around de-colonising the curriculum and sustainability and global challenges? (these may well be part of the ‘ethical’ strand of being a teacher but the argument needs to be made more strongly. And what are the other ways in which teachers manifest themselves as ethical actors?). What is the relationship between ‘Becoming a reflective practitioner’ and teacher engagement in practitioner research? (this is an assumed relationship, rather than one that is explained).  

The above comments point towards the need both to re-structure the paper in order to make the argument stronger, and to underpin the latter with a greater level of theorisation. Perhaps the problem is that the paper is trying to achieve to much – and this may be reflected in the title. ‘Bracing ourselves: embracing troubled policy change waters through a longstanding and successful university-schools teacher education partnership’ highlights the ‘brace/embrace’ concept that is explained well in the first few paragraphs but little reference is made to this framing later on in the paper. ‘Policy change waters’ introduces another metaphor which is not picked up on again. The later references to ‘strands’ and braids’ introduces another metaphor which does not seem to be aligned to the ‘brace/embrace’ ideas. And finally, the notion of ‘partnership’ does not come through that strongly in terms of the underpinning principles – what say have the schools themselves had, for example, in determining these principles?

With so much going on in the paper, things that require deeper argumentation tend to be brushed over, yet the central premise of the paper is an important one and there would be interest from readers in understanding the tensions that have arisen from government policy reform in England and the way in which one university ITE provider has sought to navigate this challenging policy landscape. So, as I have suggested above, it might be worth summarising this landscape briefly at the beginning, then discussing the challenges in light of the wider conceptualisation of what it means to be a teacher, then the principles underpinning the re-formulation of the programme for this particular ITE provider, and finally the way in which the programme has been re-conceptualised around the three ‘strands’ (and the way in which these have been theorised).

Finally, the paper is currently very England-centric. That is probably to be expected, given its focus, but it might benefit from some acknowledgement (albeit quite brief) in the introduction of the wider international context of teacher education reform, before going on to highlight that this paper is saying something important about the way in which one established and successful ITE provider has had to ‘brace’ itself against the challenges of these reforms, whilst continuing to ‘embrace’ some non-negotiable principles.

Author Response

Please see the attachment. Thank you for your review. It was very helpful.

Reviewer 2 Report

Comments and Suggestions for Authors

Thank you for the opportunity to read your paper, I found it really interesting and engaging in this difficult time of TE reform all around the world. You explained the context of TE reform really well to people living in other countries, who are familiar with TE reform generally. There are just a couple of very minor changes to make:

On page 5 lines 208-210 the font is different. 

Page 5 line 236 some more context for this website as well as consistent font size would be great. 

Overall, this was a great paper to read and I look forward to seeing it published, congratulations. 

Author Response

Thank you for your encouraging and enthusiastic review of the paper. We have made quite extensive changes in response to a more critical reading by the other reviewer. We hope you find the revised paper even more interesting and engaging! You identified a couple of very minor changes:

"On page 5 lines 208-210 the font is different". 

We have restructured the text so original page and line numbers have now changed but made sure in this submission that the font size is consistent.

"Page 5 line 236 some more context for this website as well as consistent font size would be great."

We have also addressed this point 

Thank you again for your kind response and for generously supporting the peer review process.  

Round 2

Reviewer 1 Report

Comments and Suggestions for Authors

Thank you. I really enjoyed reading the re-submission and appreciate the work that has gone into re-conceptualising the paper. As a result, this is now a much stronger submission. The introduction to the paper now sets out a clear structure with specific aims. Structuring the paper around the current policy context and then the three underpinning ‘braces’ makes the article much stronger and the argument more coherent (and less polemic in nature).

Lines 190-202: The discussion of the National Curriculum for English seems to be too much of a diversion. While it illustrates well government control, this is related to the school curriculum and not to the ITE curriculum. If the authors wanted to make the point about government control elsewhere (and, importantly, its effects) a couple of sentences, with an appropriate citation, would work. So, for example, ‘Curriculum control has also been event in relation to the school curriculum where, for example the requirements have moved from what pupils should eb able to do to what pupils should know’. What is important is not the detail here about English as a subject but rather the argument that leads to the next paragraph about how these prescriptions and requirements then determine the practice of teachers.

Overall, the issues which were somewhat problematic within the first submission have now been addressed. So, for example, the discussion of approaches to decolonisation and sustainability are now presented as exemplars of ethical approaches to the ITE curriculum, which makes the argument work more effectively.

Lines 300+ ‘Winch et al (2015) identify three distinct aspects of teachers’ professional knowledge 300 found in wider educational literature concerned with articulating the professional knowledge teachers need and which we consider necessary to being appropriately ‘informed’. It would be useful for the reader to know what these three distinct aspects are.

Figure 1 is helpful and reinforces the central argument around the three braces, making the metaphor work more effectively. The brining together of the different metaphors and concepts (brace/embracing/principles and ‘becomings’ is also now much more coherent and the conclusions pull the arguments together effectively. The reader then gets a clear sense of the principles that one provider has adopted and how these have influenced curriculum development in light of recent policy requirements. As such this makes the article a valuable contribution to the literature. Much of the recent policy critique has been around the overarching policy and the policy drivers, but this article shows us policy enactment at local level and the inherent challenges, while at the same time providing a scholarly justification for the programme’s underlying principles.

There are a few stylistic issues which need attention, for example, ‘et al’ is usually presented as ‘et al.’ (i.e. italicised). American spellings should be avoided where possible unless used consistently throughout (so ‘formalised’ rather than ‘formalized, ‘marginalised’ rather than ‘marginalized’, ‘marketised’ rather than ‘marketized’).

There is some inconsistency in the presentation of direct citations. Some are use single speech marks and others use double speech marks. This needs to be addressed for the sake of consistency and also to avoid ambiguity where some concepts are put within single speech marks but are not necessarily direct quotations. Or conversely (see line 547).

And a few specific points:

Line 31: ‘in the production’ perhaps better as ‘in the requirement for ITE providers to produce’.

Line 95: ‘the emphasis being on partnership between schools and universities,’

Line 98: ‘have not met …’ (not meet)

Line 338: ‘In’ rather than ‘On’?

Line 339: ‘op cit’, not ‘op sit’.

Line 347: ‘Let us take, as an example …’ rather than simply ‘Take …’.

Line 361: ‘…in’ not ‘…on’?

Line 439: ‘It’s just demanding’ needs to be ‘It is just demanding’ – but also needs a bit more as to how or why it is so demanding.

Comments on the Quality of English Language

The English is well presented but there are a few areas which need addressing (listed above)

Author Response

Response

We are grateful again to Reviewer 1 for their detailed engagement with our text and are confident that the article is stronger because of feedback through each stage of the review process.

Our responses include a thorough proof-read considering all the specific phrasing adjustments that Reviewer 1 suggested, alongside consistency in referencing, use of italics, punctuation within citations, use of single/double inverted commas, revising any American spellings and adding page numbers where those had been missed out.

The section on the English national curriculum (previously 190-202) was adjusted according to the guidance (including the removal from the References list of citations deleted in the revised version).

Lines 300+ where we cite Winch et al (2015) now spells out the three distinct aspects of teachers’ professional knowledge argued for in that paper.

We really like the way that you identify the contribution of this article as showing “others policy enactment at local level and the inherent challenges, while at the same time providing a scholarly justification for the programme’s underlying principles”. We’re going to share our text + that bit of feedback verbatim with other tutors on the programme and the wider PGCE Partnership. Thank you!